# Monitoring and Genotyping of Wild Grapevine (*Vitis vinifera* L. subsp. *sylvestris*) in Slovenia

**DOI:** 10.3390/plants13091234

**Published:** 2024-04-29

**Authors:** Andrej Perko, Oliver Trapp, Erika Maul, Franco Röckel, Andrej Piltaver, Stanko Vršič

**Affiliations:** 1University Centre of Viticulture and Enology Meranovo, Faculty of Agriculture and Life Sciences, University of Maribor, Pivola 10, 2311 Hoče, Slovenia; stanko.vrsic@um.si; 2Julius Kühn Institute (JKI)-Federal Research Centre of Cultivated Plants, Institute for Grapevine Breeding Geilweilerhof, 76833 Siebeldingen, Germany; oliver.trapp@julius-kuehn.de (O.T.); franco.roeckel@julius-kuehn.de (F.R.); 3Institute for the Systematics of Higher Fungi, Velika vas 17, 1262 Dol pri Ljubljani, Slovenia; anpiltaver@gmail.com

**Keywords:** wild grapevine, SSR markers, conservation, genetic fingerprinting, Slovenia

## Abstract

*Vitis vinifera* L. subsp. *sylvestris* (*sylvestris*) is the only native wild grapevine in Eurasia (Europe and western Asia) and is the existing ancestor of the grapevine varieties (for wine and table grape production) belonging to the subsp. *sativa*. In Slovenia, the prevailing opinion has been that there are no Slovenian *sylvestris* habitats. This study describes *sylvestris* in Slovenia for the first time and aims to present an overview of the locations of the wild grapevine in the country. In this project, a sample set of 89 accessions were examined using 24 SSR and 2 SSR markers plus APT3 markers to determine flower sex. The accessions were found in forests on the left bank of the Sava River in Slovenia, on the border between alluvial soils and limestone and dolomite soils, five different sites, some of which are described for the first time. The proportion of female to male accessions differed between sites. At two sites, female plants dominated; at others, the ratio was balanced. The plants’ genetic diversity and structure were compared with autochthonous and unique varieties of subsp. *sativa* from old vineyards in Slovenia and with rootstocks escaped from nature from abandoned vineyards. *Sylvestris* was clearly distinguishable from *vinifera* and the rootstocks. Based on genetic analyses, it was confirmed that Slovenian *sylvestris* is closest to the Balkan and German *sylvestris* groups. Meanwhile, a safety duplication of the wild grapevine accessions has been established at the University Centre of Viticulture and Enology Meranovo, Faculty of Agriculture and Life Sciences at the University of Maribor.

## 1. Introduction

The European wild grapevine *V. vinifera* L. subsp. *sylvestris* (C. C. Gmelin) Hegi is the progenitor of the cultivated vine *V. vinifera* L. subsp. *sativa* (DC.) Hegi. The wild subspecies survived the Ice Age in small refugia (sites with isolated or relict populations), spreading from these sites into alluvial forests [1]. The cultivated grapevine has played an important role economically and culturally for many centuries. However, its ancestor, the European wild grapevine, is threatened with extinction [2]. The surviving specimens in the wild are endangered by disease and pests such as phylloxera (*Daktulosphaira vitifoliae* Fitch), as well as by fungal diseases (e.g., downy mildew, powdery mildew). Human interventions such as deforestation and river regulation have also destroyed the habitat of the European wild grapevine. The situation further deteriorated when American species were introduced to Europe in the 19th century to control phylloxera (*Daktulosphaira vitifoliae* Fitch) [3]. Because its habitats are often located near vineyards, the European wild grapevine is endangered by hybridization with its cultivated progeny, and especially with naturalized rootstocks derived from viticulture, such as the invasive *Vitis riparia* Michx and others. Hybrids, such as Isabella (resistant to phylloxera, downy mildew, and powdery mildew), are displacing wild native grapevines from their natural habitats [4]; they are able to enter large rivers from abandoned vineyards and invade floodplain forests, where they spread as so-called neophytes (non-native species) and interbreed with native European wild grapevines. The resulting genotypes are more tolerant to diseases and phylloxera, dooming the native European wild grapevine to failure [5].

A few natural populations of European wild grapevine have survived in small, dissociated populations in remnant habitats. Examples include those in Szigetkoez (Fertő-Hanság National Park) in Hungary [3]; in Germany, those in the Upper Rhine Valley [2]; and, most probably, those along the Sava River in Slovenia, as confirmed in this study (until now, the prevailing opinion has been that there are no habitats of *sylvestris* in Slovenia). It is one of the rarest plant species in Germany and is considered critically endangered [6,7]; it is therefore strictly protected [8]. It is somewhat more common at similar sites in southeastern Europe [9].

The maintenance of existing populations is of great importance for the conservation of biodiversity, and in particular because of their role in the evolution of the vine. Wild grapevines are also especially resistant to flooding and active limestone, on account of which they are of potential use in breeding by hybridization with commercial rootstocks [10,11,12]. Due to the extremity of the situation, ex situ conservation is the means of choice for the preservation of *sylvestris*. In such a project, the last surviving specimens along the Sava River would be propagated via cuttings at UC, with an aim to return the rootlings to promising sites in the alluvial forest. To ensure natural gene flow, genetically diverse specimens would be planted together in beds. In this way, a sustainable wild grapevine population could be maintained. Meanwhile, a complete genetic copy of wild grapevine would be established at UC, and this valuable genetic resource would be harnessed for sustainable viticulture. Biodiversity conservation has a clear practical value for humanity, as some of the accessions of *sylvestris* have demonstrated a relatively high tolerance to grapevine diseases and represent a valuable genetic resource for resistance breeding [1,13]. In the present study, microsatellite markers (SSRs) were used to estimate the genetic variation in native specimens of *sylvestris* in Slovenian habitats. The main objective of this work was to assess the occurrence and genetic diversity of native wild grapevines in the territory of Slovenia and to compare this with other European *sylvestris* populations.

## 2. Results

### 2.1. Genetic Diversity of the Slovenian Sylvestris Germplasm Compared to Other Sylvestris Populations

Genetic data from 20 nuclear microsatellites drawn from 1229 genotypes of the *sylvestris* population were used to calculate genetic indices. The range of allele size (Ra), number of different alleles (Na), effective number of alleles (Ne), Shannon’s information index (I), observed heterozygosity (Ho), expected heterozygosity (He), and fixation index (F) were calculated to assess the genetic diversity of the *sylvestris* germplasm included in this study (Table 1). The genetic indices of Slovenian *sylvestris* are listed separately in the rows. The number of alleles per SSR locus ranged from 6 (VVIN16) to 29 (VVMD28) and from 3 (VVIN73) to 15 (VVMD28) in the Slovenian population (Slo pop). The total numbers of alleles (Na) were 315 and 148 (Slo pop). The numbers of effective alleles (Ne) ranged from 2.517 (VVIN73) to 8.918 (VVMD32), with an overall mean of 5.229, and in the Slo pop from 1.429 (VVMD27) to 4.349 (VVMD7), with an overall mean of 2.485. The highest Shannon’s information index was observed at the VVMD28 locus (2.487), and the lowest at the VVIN16 locus (1.181), whereas the mean value of the SSR loci was 1.883, and in the Slo pop, the highest was observed at the VVMD7 locus (1.715) and the lowest at the VVIN73 (0.539). Observed heterozygosity (Ho) was highest at VVS2 and lowest at VVMD21, ranging from 0.431 to 0.745, with an overall mean of 0.628. In the Slo pop, Ho ranged from 0.101 (VVMD21) to 0.798 (VVIP60), with a mean of 0.533. The values of expected heterozygosity (He) ranged from 0.603 (VVIN73) to 0.888 (VVMD32), with a mean of 0.782, and from 0.300 (VVMD27) to 0.770 (VVMD7) in the Slovenian population, with a mean of 0.558. The fixation index (Fst) or (F) estimate helps to determine the degree to which a group of populations differ from each other. The mean F-value for the data set was 0.199, and 0.063 for the Slo pop.

Genetic diversity indices at the population level showed that the number of alleles per locus was greatest in pop1 (Armenia) (10.550) (Table 2), with the lowest value found in pop8 (Hungary; 2.700). The Slovenian number was 7.400. The total Ne value in the data set was 3.251; the largest value was that of pop1 (5.296), and the lowest was that of Hungary, at 1.726 (pop8), while the Slovenian (pop11) value was 2.482. The observed and expected heterozygosity (Ho and He) was highest in pop1 and lowest in pop10. The fixation index (F) was positive in all populations and ranged from 0.039 (pop9) to 0.183 (pop10). In the Slovenian population (pop11), the F value was 0.063.

The genetic distance between eleven *sylvestris* populations was estimated using pairwise Fst values and Nei´s genetic distance [14] (Table 3). Based on the pairwise population matrix of Nei´s genetic distance, the greatest distance was found between the German (pop9) and Israeli (pop3) populations (1.266). The Slovenian population (pop11) was found to be closest to the Croatian (pop7) (0.080), followed by the Bosnian and Herzegovinan (pop6) (0.151), then those of Hungary (pop8) (0.154), Germany (pop9) (0.206), Spain (pop5) (0.263), Sicily (pop4) (0.294), Transcaucasia (pop2) (0.495), Armenia (pop1) (0.513), and finally Israel (pop3) (0.923). Nei´s distance for the Slovenian and other populations may be in line with geographic distance. The Fst values ranged from 0.027 for the Slovenian (pop11) and Croatian (pop7) populations to 0.314 between pop3 (Israel) and pop9 (Germany).

### 2.2. Genetic Diversity of the Slovenian Sylvestris Germplasm

Based on the allele profiles, statistical indices were calculated, and the genetic diversity of *sylvestris*, cultivars, hybrids, and rootstocks was determined to give proof of sample purity (Table 4).

The number of alleles per locus (Na) was 8.042 for the *sylvestris*, 9.000 for the cultivars, 8.083 for the hybrids, and 8.542 for the rootstock samples. The *sylvestris* samples had the lowest Ne value (2.749), and the highest Na value was found in the hybrids. The observed and expected heterozygosity (Ho and He) was highest in the hybrids and lowest in the *sylvestris* population. The fixation index (F) was negative in the cultivars (−0.056) and hybrids (−0.030), while it was positive in *sylvestris* (0.065) and the rootstocks (0.126) (Table 4).

The pairwise Nei´s genetic distance and Fst values for *sylvestris*, cultivars, hybrids, and rootstock samples are shown in Table 5. Nei´s genetic distance ranged from 1.368 (*sylvestris*—rootstock) to 0.264 (cultivars—hybrids). The Fst values confirmed the pattern, with the highest value being 0.155 (rootstock—*sylvestris*) and the lowest value, 0.032 (hybrids—cultivars).

### 2.3. Population Structure Analysis and Differentiation

The genetic diversity of the Slovenian *sylvestris*, cultivars, hybrids, and rootstocks was first assessed by DAPC analysis of SSR profiles (Figure 1a). The *sylvestris* samples formed a compact cluster with two outer layers in the upper left of the diagram. The cultivar samples formed a cluster in the lower right of the diagram. The hybrids were located between the cultivars and the rootstocks, with some samples being closer to the hybrids and some to the rootstocks.

Samples were also distinguished by PCoA analysis based on genetic distance (Figure 1b). The distribution pattern strongly resembled that of DAPC, with a clearer overlap in the *sylvestris* cluster and a dispersion of hybrid samples between cultivars and rootstocks. The two PCoA axes explained 22.73% of the observed variance. The first dimension (Axis1) explained 17.1%, while the second (Axis2) explained 5.63% of the total variation in the set.

Another method used to estimate genetic relationships between *sylvestris*, cultivars, hybrids, and rootstocks was a clustering algorithm implemented in the Structure program. The statistics of Evanno et al. [15] showed the highest probability for K = 3 (Figure 2).

### 2.4. Relationships between Slovenian and Other Sylvestris Populations

The genetic diversity of the *sylvestris* population was first assessed via DAPC analysis of the SSR profiles (Figure 3a). *Sylvestris* germplasm was divided into four groups; Israel and Spain were distinct from the others, while some Transcaucasian accessions belonged to the Slovenian, Croatian, Bosnian, German, Hungarian, Italian, and Armenian *sylvestris* groups. The distribution pattern produced by the PCoA analysis based on genetic distances (Figure 3b) closely resembled that produced by the DAPC. The projections are shown below in a two-dimensional scatter plot (Figure 3). The PCoA 2D projection of the first two principal axes accounted for 18.82% of the total observed variance.

A third independent method for assessing the relationships between genotypes was the clustering algorithm implemented in the Structure program. The statistics of Evanno et al. [15] showed the highest probability for K = 3. The simulation of the K = 3 structure divided the *sylvestris* populations into three groups (Figure 4).

The populations from Slovenia, Slovenia (Croatia), Germany, and Hungary, as well as samples from Bosnia and Herzegovina, belonged to Group 1, with Q values of more than 0.90. The Israeli population with the same Q value belonged to Group 2, and the Transcaucasian population to Group 3. The Croatian population belonged to Group 1, with a Q value of 0.79, while the Armenian population had a Q value below 0.70 and was split between Group 3 (Q value 0.669) and Group 2 (Q value 0.303). The Italian (Sicilian) population was also split between Group 2 (Q-value 0.620) and Group 1 (Q-value 0.354), while the Spanish *sylvestris* population was split between Group 1 (Q-value 0.581) and Group 2 (Q-value 0.407).

### 2.5. Flower Phenotypes

Flower phenotype analysis was performed for all genotypes collected at the *sylvestris* site (89 genotypes) and analyzed by a combination of three genetic markers: APT3, VVIB23, and Gf02–31. The Gf02–31 and APT3 markers distinguish female from hermaphrodite or male plants, while VVIB23 can identify male and female plants.

Four different allele patterns were determined at the APT3 loci: 38 genotypes showed 268/268 (F), 1 genotype 336/336 (F), and 50 genotypes 268/466 (M/H). The GF02–31 marker is highly informative, as well as understandable. Two different patterns were detected at the GF02–31 loci: 39 genotypes showed 248/248 (F) and 50 genotypes showed 248/260 (M). At the VVIB23 loci, 3 genotypes showed 288/290 (M), 35 genotypes showed 290/290 (F), 3 genotypes showed 290/304 (F), 46 genotypes showed 290/308 (M), 1 genotype showed 304/304 (F), and 1 genotype showed 304/308 (M).

The ratio between male and female plants was found to be predominantly male, except in the case of Brič, where a higher proportion of female plants was observed (Table 6). Flower phenotypes of all accessions of *V. vinifera* subsp. *sylvestris* from Slovenia (pop11, Table 7) evaluated with DNA-based flower sex markers are presented in the Appendix A.

## 3. Discussion

This is the first report to detail and assess *sylvestris* plants and their occurrence in Slovenia. *Vitis vinifera* L. subsp. *sylvestris* is the only Vitis species native to Eurasia. Its habitat is now known to include Slovenia, although it was previously assumed that wild grapevines did not exist within Slovenia, which hitherto has not been mentioned as a country of origin [22]. Most wild grapevine in Slovenia grows in privately owned forests. This can pose a problem for the conservation of accessions, as it can be very difficult to convince owners to preserve them without financial support. Landowners have sometimes not recognized the *sylvestris* plant, and in recent years, some specimens have been uprooted because they were misidentified and confused with invasive plants and climbers such as clematis (Clematis vitalba). In addition, many specimens have been destroyed by deforestation. The *sylvestris* plants discovered so far have been found on the border between limestone and alluvial soils and dolomite. Growing in an unprotected area, their existence is threatened; they are not considered endangered plants in Slovenia as they are in other European countries (e.g., Austria, Hungary, Germany, France, Spain) [23,24].

Until now, there has been no systematic genetic characterization of the Slovenian wild grapevine. Our discovery comprises 126 accessions from 5 locations, of which 93 were analyzed and 89 genotypes were selected for further investigation. Based on DAPC, PCoA, and Structure analyses, the population of Slovenian *sylvestris* can be seen to differ from cultivars and rootstocks. The results indicate a very pure population, with 95% of genotypes having a Q ≥ 0.70 and 89% having a Q ≥ 0.90. It is noteworthy that one sample from the Krnice site (sample 21) showed a similarity, at Q = 0.86, with the cultivars. Three samples (72 at the Vinje Katarija site and 2 and 7 at the Bric site) had a Q ≤ 0.70 [16], suggesting they are hybrids of *sylvestris* and *vinifera*, with a greater proportion of *sylvestris*. It has been shown that accessions in close proximity to inhabited areas, such as sample 21, taken near the village of Kovk, which has numerous grapevines on pergolas nearby, have a higher probability of contact with *vinifera*. The purest population of *sylvestris* was found at the Krnice site in the Hrastnik area, on the left bank of the Sava. Here, 28 out of 30 genotypes exhibited Q ≥ 0.99, while one remained at Q ≥ 0.91. The presence of *sylvestris* in the vicinity of inhabited areas is an indication of an increased risk of deterioration of genetic potential or contact with *vinifera*. The most populous site examined, Vinje-Katarija, is home to two subgroups, one grouped around Vinje and the other around Katarija. Within the Vinje-Katarija locality, 60 accessions were identified, of which 51 were analyzed. Of these, 88% of the genotypes exhibited a Q ≥ 0.95, 4% a Q ≥ 0.84, and 6% a Q ≥ 0.76, while sample 72 represents a hybrid, with approximately equal proportions of *sylvestris* and *vinifera*. *Sylvestris* plants were found to be most common in the village of Katarija. Despite their dense concentration, they are all genetically different and are not vegetative offspring. Four plants were discovered at the Rašica site, but three of these were destroyed. An accession site that is isolated from other sites is Žusem. In western Slovenia, on the border with Croatia along the Dragonja River, is the Brič site, where seven plants were found. Of these, four had a Q ≥ 0.96, one a Q ≥ 0.71 and two a Q ≤ 0.70, but with a *sylvestris* proportion of at least 0.65.

Among the Vitis vinifera varieties analyzed, ‘Gewuerztraminer’, ‘Chasselas’, ‘Merlot’, and ‘Pinot’ had the highest proportions of *sylvestris* based on our marker set and samples. ‘Gewuerztraminer’ could be regarded as a hybrid (Q: *vinifera* = 0.615, *sylvestris* = 0.318), while ‘Chasselas Blanc’ also contained a significant proportion of *sylvestris* (Q: *vinifera* = 0.742, *sylvestris* = 0.256), as has been previously reported [17,20,21].

This study also performed a comparison of the Slovenian wild population with previously published datasets produced by other authors [16,17,18,19,20,21]. Our comparison extended to wild accessions from neighboring countries (Italy, Croatia, Hungary), Bosnia and Herzegovina (Balkans), Germany (Central Europe), Spain (Iberian Peninsula), Israel, Transcaucasia, and Armenia. A significant differentiation within and between the populations was detected by DPCA, PCoA, and Structure analysis, and Fst values similarly revealed differentiation. *Sylvestris* showed a clear sub-division into two main groups: West and East *sylvestris*. Fst values further emphasized the heterogeneity between populations. The Slovenian population showed a close genetic relationship with Croatian, Bosnian, Hungarian, and German populations, which is consistent with the results from [21]. Strikingly different populations, however, were found to occur in Israel and Transcaucasia. In addition, Spanish and Italian populations showed hybrid characteristics, representing a mix of Balkan and Eastern European populations. These results confirm previous research on the evolution of *sylvestris* subgroups and suggest that the Slovenian population most likely belongs to the Balkan *sylvestris* subgroup [25].

Preserving the diversity of wild grapevine through ex situ conservation is crucial in preventing the extinction of these invaluable accessions and safeguarding their significant potential for future breeding efforts. Several accessions have been vegetatively propagated and planted at UC Meranovo to ensure uniform growing conditions for their subsequent detailed morphological characterization [26].

## 4. Materials and Methods

### 4.1. Plant Material and Study Site

Samples of *sylvestris* (126 plants) were found at five different sites (Figure 5) in floodplain forests in Slovenia. The accessions were found in forests on the orographic left bank of the Sava River in Slovenia; on the border between alluvial soils and limestone and dolomite at three different sites (nos. 1, 2, and 3; Figure 6); at one limestone site near the border with Croatia on the right bank of the Dragonja River (no. 5; Figure 6); and at one dolomite site in the village of Žusem (no. 4; Figure 6), 25 km from Celje. All habitats were described for the first time. The *sylvestris* plants were identified by molecular identification, and potential *sylvestris* plants were recognized using the OIV descriptors for morphological assessment of *Vitis sylvestris* [26]. A total of 1229 *sylvestris* samples comprised 1140 samples of *sylvestris* germplasm from Armenia (pop1), Transcaucasia (Armenia, Azerbaijan and Georgia) (pop2) [17], Israel (pop3) [18], Italy (Sicily) (pop4) [19], Spain (pop5) [20], Bosnia and Herzegovina (pop6), Croatia (pop7), Hungary (pop8), and Germany (pop9). In addition, the few Slovenian samples in pop10 [21] indicate that populations of *sylvestris* still exist in Slovenia. In this work, 89 new accessions of *sylvestris* from Slovenia (pop11) were genotyped. Slovenian samples were also compared with cultivars, hybrids, and rootstocks which had escaped to nature from abandoned vineyards.

### 4.2. DNA Extraction and Microsatellite (nSSR) Analysis

Total genomic DNA was extracted from young leaves, which were ground with the MM 300 Mixer Mill System (Retsch, Haan, Germany) and stored at −80 °C prior to use. Grapevine DNA was extracted using the NucleoSpin Plant II Kit (Macherey-Nagel, Düren, Germany). The extracted DNA was quantified by spectrometry and diluted to a concentration of 1 ng/µL. Microsatellite fingerprinting of genotypes was performed at 24 microsatellite loci: VVS2, VVMD5, VVMD7, VVMD21, VVMD24, VVMD25, VVMD27, VVMD28, VVMD32, VrZAG62, VrZAG67, VrZAG79, VrZAG83, VMC4f3.1, VMC1b11, VVIb01, VVIn16, VVIh54, VVIn73, VVIp31, VVIp60, VVIv37, VVIv67, and VVIq52 [27,28,29,30,31,32,33,34]. All forward primers were 5’ end-labeled with fluorescent dyes (FAM, HEX, TAMRA, ROX, and PET).

Fragment length was determined by capillary electrophoresis using the ABI 3130xl Genetic Analyzer (Applied Biosystems, Life Technologies, Waltham, MA, USA). The combinations of microsatellite loci (multiplexes) were optimized in the laboratory of the Julius Kuehn Institute. The use of different markers and different fragment lengths allowed for multiplexing of polymerase chain reactions (PCR) with up to five SSR markers characterized by similar annealing temperatures. First, 1 ng of DNA was mixed with 2× KAPA2G Fast PCR Kit (Duren, Germany) to prepare 5 µL reaction mixes containing a master mix and 100 pmol of each primer. Amplification was performed in ABI 9700 thermal cyclers (Applied Biosystems, Foster City, CA, USA), starting with an initial denaturation of 3 min at 95 °C, followed by 30 cycles of denaturation at 95 °C (15 s), annealing at 60 °C (30 s), and extension at 72 °C (30 s). A final extension was performed at 72 °C for 7 min. Then, 1 µL of PCR product was used to determine the fragment length, and results were processed using GeneMapper 5.0 software (Applied Biosystems, Foster City, CA, USA) based on a fluorescently labeled size marker ranging from 75 to 500 bp [34]. To correct for amplification shifts between different multiplexes, SSR profiles were adjusted by adding DNA from standard cultivars from the Julius Kuehn Institute laboratory, ‘Muscat a petit grains’ and ‘Cabernet franc’, to each PCR amplification run.

### 4.3. Flower Phenotype Analysis

One of the most important distinguishing features between cultivated and wild grapevines is the flower morphology. Cultivated varieties of *Vitis vinifera* L. subsp. *vinifera* usually have hermaphrodite flowers, whereas almost all wild *Vitis* species are dioecious, with separate male and female individuals. Therefore, flower sex determination was performed on all 89 genotypes collected from *sylvestris* sites (pop11), based on the available genetic methods. A combination of three different genetic markers was used to analyze the flower phenotype: the APT3 marker, developed from the adenine phosphoribosyltransferase gene; the marker VVIB23, recommended by several authors [35] as a marker for grapevine flower sex; and GF02–31 [36]. Since flower sex is an important trait in proving the authenticity of *sylvestris*, all three markers were included in order for them to complement each other in sex determination.

### 4.4. Genetic Diversity Analysis

Different measures were employed to assess genetic variability between 1229 *sylvestris* genotypes at 20 SSR loci and 149 genotypes at 24 SSR loci, encompassing Slovenian *sylvestris*, cultivars, hybrids, and rootstocks. The number of distinct alleles per locus (Na), number of effective alleles (Ne), observed heterozygosity (Ho), expected heterozygosity (He), Shannon’s information index (I), and fixation index (F) were calculated for each locus for wild grapevine and other profiles used for comparison. GenAlEx software, version 6.5, was used to calculate genetic diversity statistics for each locus [37,38].

Genetic relationships among accessions were assessed via distance-based cluster analysis using the neighbor-joining method (NJ), as implemented in MEGA 11.0 software.

### 4.5. Population Structure

Cluster analysis based on the Bayesian model was performed using the Structure V2.3.4 software package [39] to determine the optimum genetically supported groupings based on microsatellite data. The Structure configuration was set to use an admixture model and independent allele frequencies from the population. The allele frequency parameter (lambda) was set to 1, as suggested in the Structure manual. Different numbers of putative populations (K) were tested, ranging from 1 to 10. The burning time and number of replicates (MCMC) were set to 25,000 and 25,000, respectively, for each independent run, each having 10 iterations. The choice of the most likely number of clusters (best K) was assessed using the ad hoc delta K statistic, as described in [15], with the Structure harvester [40].

The genetic relationships between the investigated genotypes were analyzed using discriminant analysis of principal components (DAPC), implemented in R/adegenet [41]. Principal coordinate analysis (PCA) was used to indicate genetic divergence between samples in a multidimensional space over a distance matrix with data standardization using GenAlEx software, version 6.5 [37,38].

## 5. Conclusions

The Slovenian wild grapevine population studied in this project was found to be different from rootstocks and *vinifera* varieties, which is consistent with previous research results. The observed closer relationship to rootstocks or *vinifera* varieties in some accessions likely relates to hybridization between wild grapevine and cultivated varieties. Confirmation of the wild grapevine *Vitis vinifera* subsp. *sylvestris* in Slovenia is still pending, although extensive monitoring was carried out in alluvial forests between 2019 and 2022. Wild grapevine populations thrive mainly on dolomite and limestone soils, especially along the Sava River at three main sites and at two smaller sites along the southwestern border, near the Dragonja River in eastern Slovenia. Genetic analysis of 89 accessions revealed a balanced ratio between female and male plants, although at two sites, Vinje-Katarija and Brič, female plants were more strongly represented.

Urgent action and ongoing research are essential to protect the unique Slovenian wild grapevine population and curb genetic depletion. The conservation of this population is particularly important in light of the fact that certain *sylvestris* specimens show resistance to grapevine diseases and thus represent a valuable genetic reservoir for resistance breeding. It is also apparent that *sylvestris* can inherit traits relevant to adaptation to different climates. In order to preserve the *sylvestris* population, it is suggested that suitable sites be selected in the national parks to which vegetative progeny can be transplanted in order to maintain their different genotypes. Meanwhile, a comprehensive genetic replication of the wild grapevine has been established at the Meranovo University Centre, part of the Faculty of Agriculture and Life Sciences at the University of Maribor.

## Figures and Tables

**Figure 1 plants-13-01234-f001:**
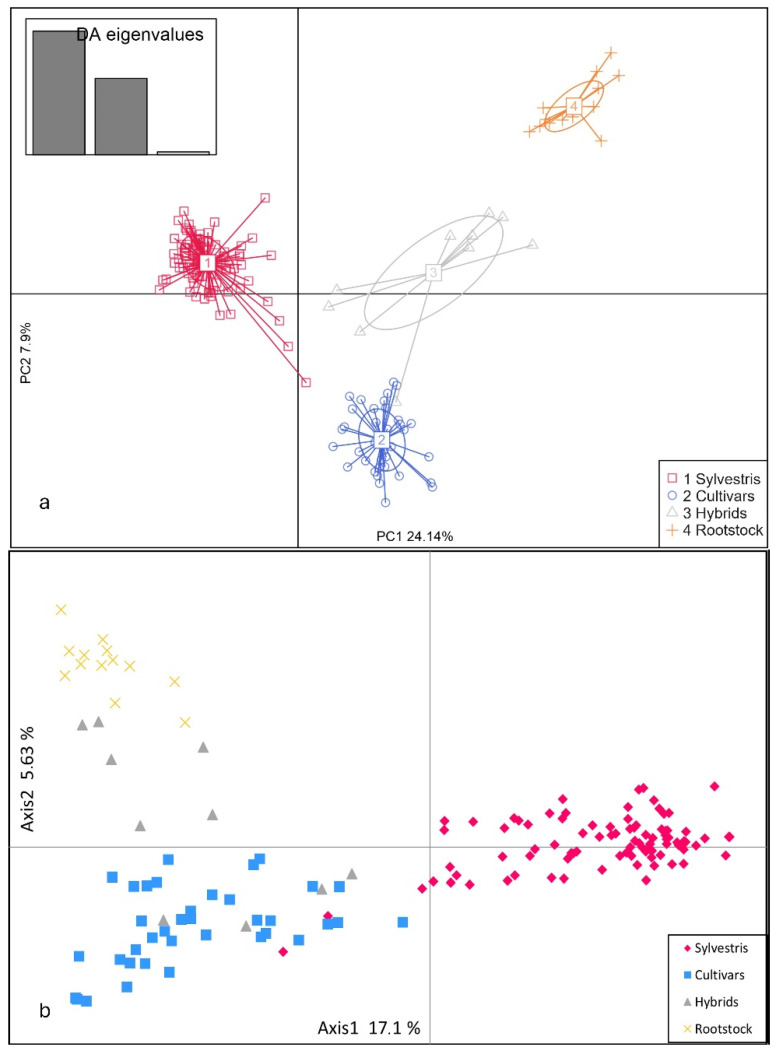
Analysis of Slovenian *sylvestris* germplasm, cultivars, hybrids, and rootstocks. Discriminant analysis of principal components (DAPC) (**a**) and principal component analysis (PCoA) (**b**) of the 149 *sylvestris*, cultivar, hybrid, and rootstock samples represented by two axes, using a covariance matrix of 24 SSR loci.

**Figure 2 plants-13-01234-f002:**
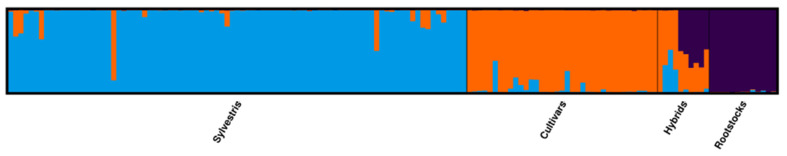
Barplot displaying the admixture proportions of 149 genotypes (1 = *sylvestris*, 2 = cultivars, 3 = hybrids, 4 = rootstocks) as estimated by Structure analysis at K = 3. Each accession is represented by a single vertical bar, presented in K colors.

**Figure 3 plants-13-01234-f003:**
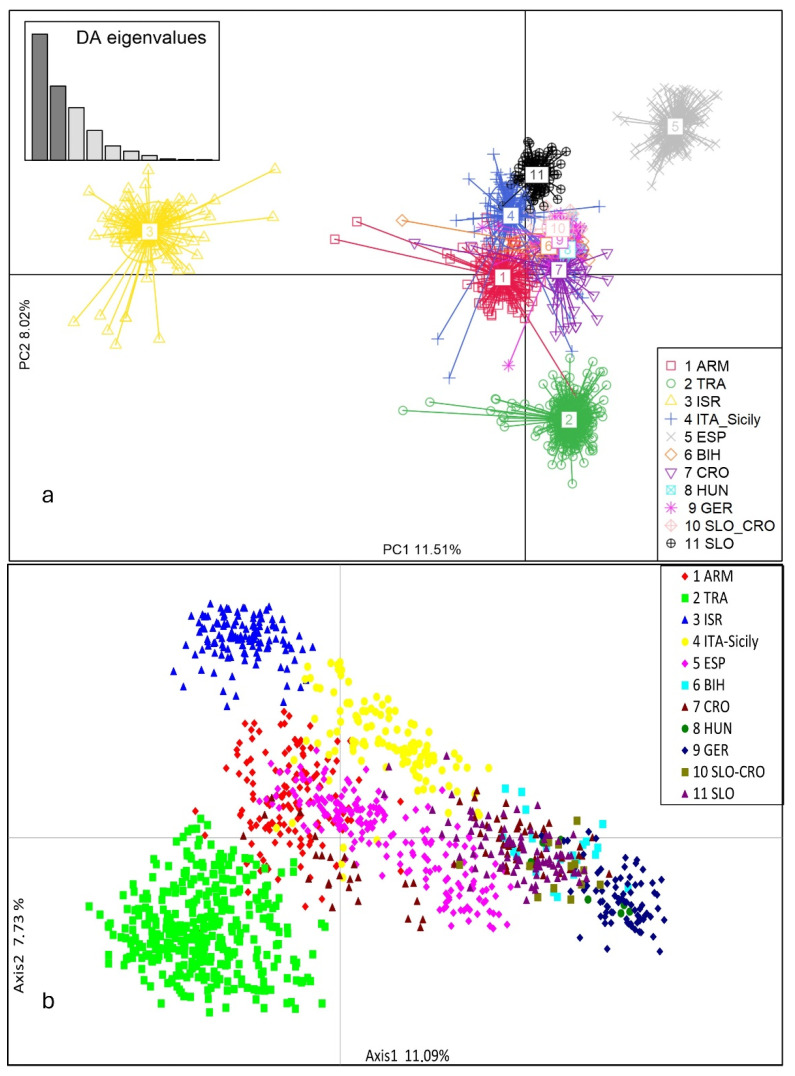
Analysis of the *sylvestris* germplasm. Discriminant analysis of principal components (DAPC) (**a**) and principal component analysis (PCoA) (**b**) of the 1229 samples are represented by two axes, using a covariance matrix of 20 SSR loci.

**Figure 4 plants-13-01234-f004:**
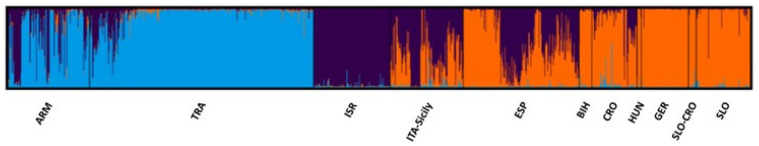
Barplot displaying the admixture proportions of 1229 *sylvestris* genotypes as estimated by Structure analysis at K = 3. (ARM = Armenia, TRA = Transcaucasia, ISR = Israel, ITA = Italy (Sicily), ESP = Spain, BIH = Bosnia and Herzegovina, CRO = Croatia, HUN = Hungary, GER = Germany, SLO-CRO = Slovenia–Croatia, SLO = Slovenia).

**Figure 5 plants-13-01234-f005:**
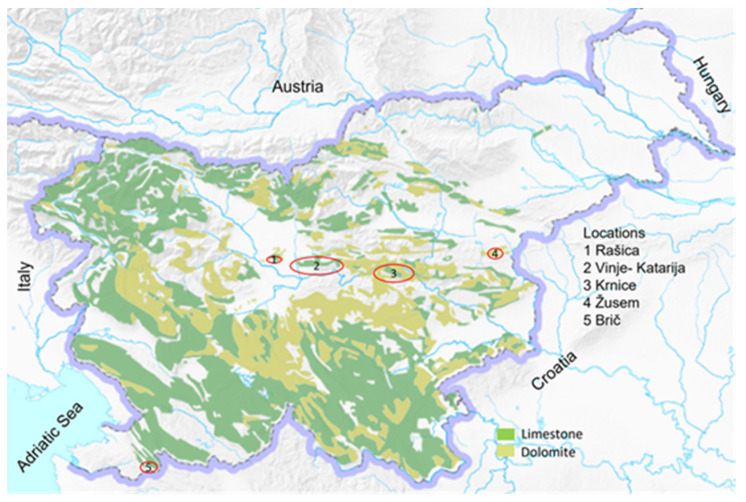
Locations of *sylvestris* in Slovenia.

**Figure 6 plants-13-01234-f006:**
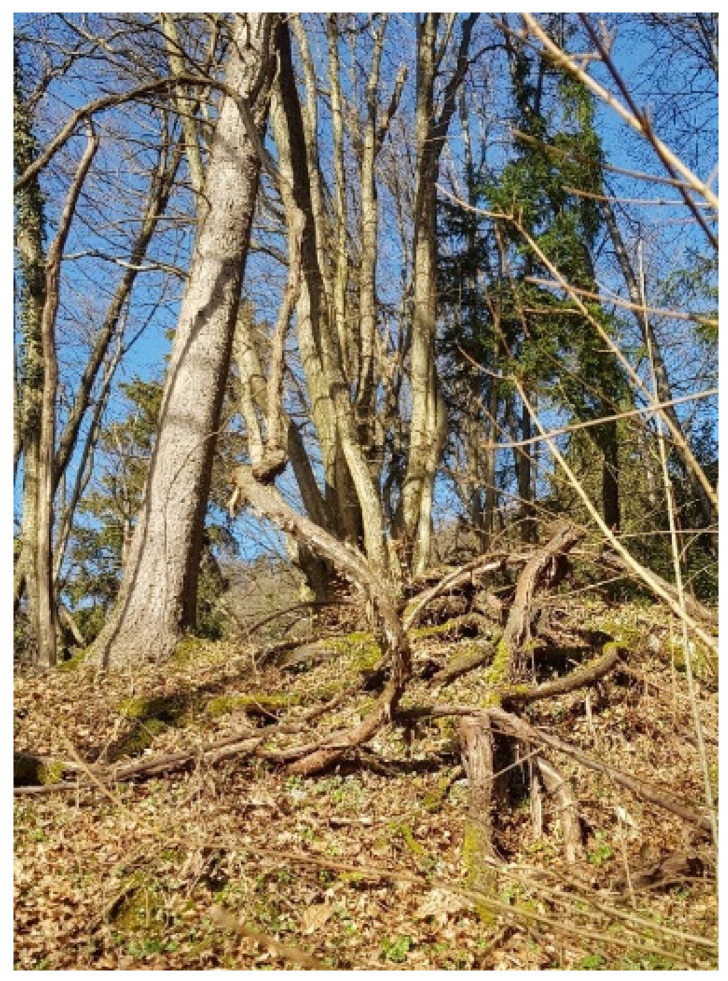
The largest and most established known vine of *sylvestris* in Slovenia (male plant, trunk circumference = 59 cm in 2021, with 4 branches more than 40 m long), found in Vinje.

**Table 1 plants-13-01234-t001:** Descriptive statistics and genetic diversity at 20 microsatellite loci of 1229 *sylvestris* genotypes, including 89 genotypes from Slovenia.

Locus		Ra (bp)	N	Na	Ne	I	Ho	He	F
VVS2	Total	123–165	1207	18	8.121	2.295	0.745	0.877	0.151
	Slovenian	133–157	89	9	3.899	1.524	0.764	0.744	−0.028
VVMD5	Total	224–250	1080	14	6.350	2.061	0.667	0.843	0.209
	Slovenian	228–242	89	6	2.387	1.032	0.584	0.581	−0.005
VVMD7	Total	231–267	1219	19	6.397	2.174	0.742	0.844	0.121
	Slovenian	231–265	89	11	4.349	1.715	0.742	0.770	0.037
VVMD25	Total	237–273	1205	18	5.738	1.988	0.676	0.826	0.182
	Slovenian	239–267	89	6	2.723	1.163	0.618	0.633	0.023
VVMD27	Total	176–198	1202	12	4.343	1.828	0.611	0.770	0.207
	Slovenian	176–192	89	8	1.429	0.702	0.292	0.300	0.027
VVMD28	Total	218–282	1194	29	7.640	2.487	0.695	0.869	0.200
	Slovenian	228–272	89	15	3.161	1.709	0.652	0.684	0.047
VVMD32	Total	234–292	1033	20	8.918	2.408	0.701	0.888	0.211
	Slovenian	238–272	89	6	2.052	1.025	0.483	0.513	0.057
VRZAG62	Total	184–224	814	12	3.984	1.693	0.624	0.749	0.167
	Slovenian	186–224	89	8	1.802	0.934	0.348	0.445	0.218
VRZAG79	Total	233–261	629	15	3.810	1.786	0.563	0.738	0.237
	Slovenian	233–259	89	10	2.175	1.210	0.461	0.540	0.147
VVIN16	Total	145–157	1225	6	2.749	1.181	0.518	0.636	0.187
	Slovenian	145–157	89	5	2.914	1.130	0.708	0.657	−0.078
VVIN73	Total	248–272	1096	9	2.517	1.237	0.497	0.603	0.175
	Slovenian	260–268	89	3	1.471	0.539	0.303	0.320	0.052
VVIP60	Total	276–332	914	17	4.727	1.856	0.644	0.788	0.183
	Slovenian	306–326	89	7	3.560	1.449	0.798	0.719	−0.109
VVMD24	Total	202–220	1213	10	4.361	1.678	0.671	0.771	0.129
	Slovenian	204–214	89	6	1.749	0.870	0.449	0.428	−0.050
VVMD21	Total	231–276	1192	14	3.609	1.587	0.431	0.723	0.403
	Slovenian	244–257	89	6	1.590	0.766	0.101	0.371	0.727
VVIB01	Total	269–319	1202	13	3.265	1.390	0.549	0.694	0.208
	Slovenian	269–295	89	4	2.138	0.832	0.539	0.532	−0.013
VVIH54	Total	133–181	1035	23	7.362	2.307	0.700	0.864	0.191
	Slovenian	149–179	89	11	2.445	1.258	0.562	0.591	0.049
VVIQ52	Total	68–88	1214	11	3.063	1.288	0.523	0.674	0.223
	Slovenian	70–88	89	8	1.999	1.027	0.461	0.500	0.078
VVIV37	Total	142–180	1062	18	4.961	2.064	0.592	0.798	0.258
	Slovenian	150–168	89	5	2.431	1.158	0.551	0.589	0.065
VMC1B11	Total	163–203	1203	18	5.920	2.109	0.662	0.831	0.204
	Slovenian	167–195	89	5	2.586	1.135	0.607	0.613	0.011
VVIP31	Total	157–216	1215	19	6.741	2.244	0.742	0.852	0.128
	Slovenian	175–195	89	9	2.775	1.258	0.640	0.640	−0.001
Total	Total			315					
	Slovenian			148					
Min	Total			6	2.517	1.181	0.431	0.603	0.121
	Slovenian			3	1.429	0.539	0.101	0.300	−0.109
Max	Total			29	8.918	2.487	0.745	0.888	0.403
	Slovenian			15	4.349	1.715	0.798	0.770	0.727
Mean	Total		1107.7	15.75	5.229	1.883	0.628	0.782	0.199
	Slovenian		89	7.400	2.485	1.122	0.533	0.558	0.063

Na: number of different alleles; Ne: effective alleles; I: Shannon´s information index; Ho: observed heterozygosity; He: expected heterozygosity; F: fixation index.

**Table 2 plants-13-01234-t002:** Genetic diversity estimates for each population of *sylvestris* analyzed.

Population		N	Na	Ne	Ho	He	F
Pop1	Mean	132.5	10.550	5.296	0.721	0.775	0.068
	SE	0.5	0.769	0.436	0.022	0.025	0.014
Pop2	Mean	318.1	8.800	3.522	0.571	0.647	0.120
	SE	25.6	1.017	0.370	0.050	0.053	0.018
Pop3	Mean	104.8	9.300	4.458	0.595	0.647	0.079
	SE	10.1	1.212	0.644	0.063	0.067	0.021
Pop4	Mean	111.4	9.100	4.155	0.652	0.696	0.065
	SE	5.9	0.915	0.394	0.047	0.046	0.023
Pop5	Mean	182.4	8.650	4.173	0.607	0.709	0.150
	SE	9.6	0.898	0.408	0.042	0.042	0.024
Pop6	Mean	18.5	4.850	2.467	0.517	0.545	0.054
	SE	1.4	0.443	0.250	0.052	0.048	0.037
Pop7	Mean	65.3	8.250	3.605	0.588	0.683	0.142
	SE	3.4	0.523	0.315	0.030	0.027	0.023
Pop8	Mean	6.3	2.700	1.726	0.386	0.381	0.040
	SE	0.5	0.272	0.199	0.060	0.046	0.089
Pop9	Mean	68.1	4.000	1.791	0.373	0.388	0.031
	SE	5.3	0.441	0.200	0.053	0.053	0.026
Pop10	Mean	11.4	3.450	2.090	0.368	0.459	0.183
	SE	0.9	0.380	0.232	0.055	0.055	0.068
Pop11	Mean	89.0	7.400	2.482	0.533	0.558	0.063
	SE	0.0	0.638	0.178	0.039	0.030	0.039
Total	Mean	100.7	7.005	3.251	0.537	0.590	0.090
	SE	6.4	0.280	0.130	0.016	0.016	0.012

Na: number of different alleles; Ne: effective alleles; Ho: observed heterozygosity; He: expected heterozygosity; F: fixation index.

**Table 3 plants-13-01234-t003:** Estimates of pairwise population Fst values (below the diagonal) and Nei´s genetic distance of the pairwise population matrix (above the diagonal) for *sylvestris* populations.

	Pop1	Pop2	Pop3	Pop4	Pop5	Pop6	Pop7	Pop8	Pop9	Pop10	Pop11
Pop1		0.168	0.493	0.289	0.466	0.582	0.446	0.607	0.779	0.571	0.513
Pop2	**0.075**		0.723	0.495	0.483	0.572	0.484	0.527	0.624	0.546	0.495
Pop3	**0.130**	**0.204**		0.437	0.708	1.028	0.760	1.062	1.226	1.099	0.923
Pop4	**0.065**	**0.143**	**0.158**		0.301	0.365	0.216	0.450	0.571	0.412	0.294
Pop5	**0.081**	**0.136**	**0.175**	**0.095**		0.311	0.260	0.381	0.333	0.419	0.263
pop6	**0.139**	**0.198**	**0.249**	**0.143**	**0.136**		0.112	0.202	0.156	0.177	0.151
pop7	**0.066**	**0.128**	**0.171**	**0.064**	**0.072**	**0.078**		0.225	0.231	0.178	0.080
pop8	**0.186**	**0.238**	**0.304**	**0.193**	**0.184**	**0.172**	**0.127**		0.155	0.164	0.154
Pop9	**0.205**	**0.253**	**0.314**	**0.215**	**0.180**	**0.164**	**0.132**	**0.171**		0.228	0.206
Pop10	**0.162**	**0.221**	**0.283**	**0.171**	**0.173**	**0.156**	**0.105**	**0.166**	**0.192**		0.153
Pop11	**0.099**	**0.155**	**0.214**	**0.096**	**0.092**	**0.098**	**0.027**	**0.112**	**0.133**	**0.105**	

In bold, significant Fst values with *p* ≤ 0.01 calculated over 999 permutations.

**Table 4 plants-13-01234-t004:** Genetic diversity estimates for each population analyzed.

Population		N	Na	Ne	Ho	He	F
*Sylvestris*	Mean	88.917	8.042	2.749	0.553	0.585	0.065
	SE	0.083	0.669	0.222	0.035	0.030	0.035
Cultivars	Mean	37.000	9.000	4.894	0.804	0.760	−0.056
	SE	0.000	0.640	0.379	0.027	0.022	0.017
Hybrids	Mean	9.958	8.083	5.769	0.827	0.799	−0.030
	SE	0.042	0.496	0.359	0.031	0.023	0.018
Rootstocks	Mean	12.667	8.542	5.408	0.694	0.766	0.126
	SE	0.253	0.637	0.441	0.057	0.031	0.064
Total	Mean	37.135	8.417	4.705	0.720	0.728	0.026
	SE	3.253	0.305	0.214	0.022	0.016	0.020

Na: number of different alleles; Ne: effective alleles; Ho: observed heterozygosity; He: expected heterozygosity; F: fixation index.

**Table 5 plants-13-01234-t005:** Estimates of pairwise population Fst values (below the diagonal) and Nei´s genetic distance of the pairwise population matrix (above the diagonal).

	*Sylvestris*	Cultivars	Hybrids	Rootstocks
*Sylvestris*		0.492	0.547	1.368
Cultivars	**0.090**		0.264	1.074
Hybrids	**0.092**	**0.032**		0.536
Rootstocks	**0.155**	**0.093**	**0.055**	

In bold, significant Fst values with *p* ≤ 0.01 calculated over 999 permutations.

**Table 6 plants-13-01234-t006:** List of accessions and analyzed samples of *sylvestris* founded at different sites in Slovenia.

Location	Nr. of Accessions	Analyzed Samples	Nr. of Female	Nr. of Male
Rašica	1	1	0	1
Vinje-Katarija	60	51	23	28
Krnice	57	29	12	17
Žusem	1	1	0	1
Brič	7	7	4	3
Total	126	89	39	50

**Table 7 plants-13-01234-t007:** List of wild accessions of *Vitis vinifera* (1229) grouped according to their geographic location.

Country	Population Name	Sample Number	Data Source
Armenia	Population 1 (pop1)	135	Margaryan et al. (2023) [16]
Transcaucasia	Population 2 (pop2)	371	Riaz et al. (2018) [17]
Israel	Population 3 (pop3)	127	Rahimi et al. (2021) [18]
Italy (Sicily)	Population 4 (pop4)	121	De Michele et al. (2019) [19]
Spain	Population 5 (pop5)	192	De Andrés et al. (2012) [20]
Bosnia and Herzegovina	Population 6 (pop6)	21	Zdunić et al. (2020) [21]
Croatia	Population 7 (pop7)	75	Zdunić et al. (2020) [21]
Hungary	Population 8 (pop8)	7	Zdunić et al. (2020) [21]
Germany	Population 9 (pop9)	78	Zdunić et al. (2020) [21]
Croatia-Slovenia	Population 10 (pop10)	13	Zdunić et al. (2020) [21]
Slovenia	Population 11 (pop11)	89	
Total		1229	

## Data Availability

Data are available from the authors.

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
