# Peer review of "Monitoring and Genotyping of Wild Grapevine (Vitis vinifera L. subsp. sylvestris) in Slovenia"

_plants, 2024, doi:10.3390/plants13091234_

Round 1
Reviewer 1 Report
Comments and Suggestions for Authors
The MS describes the study and genotyping of wild vine (Vitis vinifera subsp. sylvestris) in Slovenia. It was previously assumed that there were no habitats for this species in Slovenia, but the study refuted this assumption. The study is limited only to the territory of Slovenia. Although of interest within local botany and conservation, such geographic specificity may be less attractive to the broader scientific community and international readers. While the research has implications for biodiversity conservation, the specific practical implications of the findings are not fully understood. This may reduce interest in the article among a wider audience.
Notes: The main note is that the MS does not disclose statistical methods, which makes the results obtained questionable. It is necessary to describe this in as much detail as possible.
It is necessary to accurately and specifically indicate which organ was used to obtain DNA, what age it was and at what height of the stem.
The authors indicate that in the studied areas there are plants with different sexes, but it is unclear what this means and what it indicates, what environmental factors may influence this. It is not clear whether the sex of a plant can change during the later stages of the growing season. It is also unclear how correct it is to compare female and male plants with each other, or whether the authors did not do this at all. This is unclear from the article.
The MS contains a lot of tabular data, which may make the main results and conclusions difficult to read and understand. The discussion of results and conclusions need more depth and analysis to better connect the study's findings to current knowledge in the field and highlight their uniqueness and significance.
Comments on the Quality of English LanguageModerate editing of English language required.
Author Response
Dear Reviewer
we are grateful for your suggestions and review. We tried to explain questions and improve manuscript. Thank you.
Kind regards
Andrej Perko

Reviewer 2 Report
Comments and Suggestions for Authors
Perko et al. report on their genetic diversity analysis on Vitis vinifera ssp. sylvestris in Slovenia. The research is of interest to better understand the extant distribution of this taxon and adds interesting data to our understanding of the population structure of the subspecies in Central Europe.
However, some flaws are evident in the manuscript that need to be fixed before publication:
1. How was the taxon identified in the field? Which morphological characters were used? At least give a citation for the field detection, i.e. on which basis you have identified individuals of ssp. sylvestris in the first place.
2. "sylvestris" should always be spelled with non-capital first letter and slanted font throughout the ms.
3. Did you morphogically check for the sex of the plant - or only used the available genetic methods? This needs to made clear to better understand the data obtained.
4. In the abstract "and 2 SSR markers plus APT3 marker" - should be "and 2 SSR markers plus APT3 markers".
5. species/subspecies names always should be in slanted font, e.g. "vinifera" in the abstract.
6. In the introduction you list a number of pests, please provide references.
7. In Table 4 you have the "Shannon-Index" in the legend, but no values are provided in the table.
8. page 7, 3rd paragraph, last sentence misses the verb of the sentence.
9. How where genotypes of the newly sampled individuals aligned to the previous collections from other sources? Did you have reference individuals? Or how was this done?
10. Page 10, line 3: add space between "and" and "Gf0231".
11. Page 10, discussion first paragraph - why could private ownership represent a problem? Please give some explanation.
12. Page 11, 3rd paragraph, should be "detected" instead of "deteched".
13. Page 14. Under 4.4, first paragraph: where did the 1229 samples come from - a previous collection or publication? Please explain and give a reference when applicable.
14. Under 4.5, line 6: should be "burnin" not "burning".
Comments on the Quality of English Language
The English is OK, except for some mistakes indicated above; a final spell check by a native speaker would still be good.
Author Response

(The authors gave the same response as above.)

Reviewer 3 Report
Comments and Suggestions for Authors
Author Response

(The authors gave the same response as above.)

Round 2
Reviewer 1 Report
Comments and Suggestions for Authors
The authors revised the manuscript and responded to most comments.
Reviewer 2 Report
Comments and Suggestions for Authors
All requests and corrections have been dealt with, I agree with the manuscript in its present form.